# Molecular and Pathological Profiling of Corresponding Treatment-Naïve and Neoadjuvant Pazopanib-Treated High-Risk Soft Tissue Sarcoma Samples of the GISG-04/NOPASS Study

**DOI:** 10.3390/biology10070639

**Published:** 2021-07-09

**Authors:** Timo Gaiser, Christian Sauer, Alexander Marx, Jens Jakob, Bernd Kasper, Peter Hohenberger, Daniela Hirsch, Ulrich Ronellenfitsch

**Affiliations:** 1Institute of Pathology, University Medical Centre Mannheim, University of Heidelberg, 68305 Mannheim, Germany; christian.sauer@umm.de (C.S.); alexander.marx@umm.de (A.M.); daniela.hirsch@umm.de (D.H.); 2Department of General, Visceral and Child Surgery, University Medical Center Göttingen, 37075 Göttingen, Germany; jens.jakob@med.uni-goettingen.de; 3Sarcoma Unit, Interdisciplinary Tumor Center Mannheim, Mannheim University Medical Center, 68305 Mannheim, Germany; bernd.kasper@umm.de; 4Division of Surgical Oncology and Thoracic Surgery, University Medical Center Mannheim, 68167 Mannheim, Germany; peter.hohenberger@umm.de; 5Department of Abdominal, Vascular, and Endocrine Surgery, University Hospital Halle (Saale), 06120 Halle (Saale), Germany; ulrich.ronellenfitsch@uk-halle.de

**Keywords:** soft tissue sarcoma, STS, pazopanib, next generation sequencing, NGS, CDK4, MAP2K1 E203K, BRAF V600E

## Abstract

**Simple Summary:**

Patients with malignant soft tissue tumors, called soft tissue sarcoma (STS), show alterations in their deoxyribonucleic acid (DNA). Pathologists use these alterations for classification of STS and as drug-related biomarkers. Some drugs show better effectiveness in association with certain genetic alterations. In this study, we examined STS tumor tissue from a specific sarcoma study (GISG-04/NOPASS) with massively parallel DNA sequencing in order to find genetic biomarkers for pazopanib, a multi-target tyrosine kinase inhibitor approved for the treatment of advanced STS. While we could not clearly identify a specific genetic target, we were able to improve diagnostic accuracy and could detect mutations that are potentially useful for individualized therapy.

**Abstract:**

In the framework of the German Interdisciplinary Sarcoma Group GISG-04/NOPASS trial, we evaluated soft tissue sarcoma samples taken before and after neoadjuvant pazopanib therapy using histopathology and next generation sequencing (NGS) to find potential predictive biomarkers. We also aimed to improve the genetically based sarcoma classification and to elucidate additional potentially druggable mutations. In total, 30 tumor samples from 18 patients consisting of 12 pre-therapeutic biopsies and 18 resection specimens following neoadjuvant pazopanib therapy were available for analyses. NGS was performed with the Oncomine Focus Assay (Ion Torrent) covering 0.03 Mb of DNA and enabled the detection of genetic variants in 52 cancer-relevant genes. Pathological analysis showed significant regression (≥50%) after pazopanib treatment in only one undifferentiated (pleomorphic) sarcoma. NGS analyses revealed a very high frequency of *CDK4* amplification (88%; 7/8) in the group of dedifferentiated liposarcoma. In addition, two potentially druggable mutations, a *MAP2K1* missense mutation (E203K) and a *BRAF* missense mutation (V600E), were traceable in two undifferentiated (pleomorphic) sarcoma patients (11%; 2/18). Our findings demonstrate that NGS testing is a powerful technology helping to improve diagnostic accuracy and offering some patients the chance for personalized medicine even in a “mutation unlikely” cohort like STS.

## 1. Introduction

Sarcomas are a heterogeneous group of malignancies descending from a mesenchymal cell of origin and are often associated with a poor prognosis. Locally limited soft tissue sarcoma (STS) can be treated curatively only with radical surgery [1]. However, the procedure is often complicated by large tumor size, extensive tumor vasculature or locally destructive growth. Therefore, pre-operative treatment strategies with the aim of tumor reduction and devascularization are of great benefit. Following this rationale, we carried out a phase II window-of-opportunity trial under the auspices of the German Interdisciplinary Sarcoma Group (GISG-04/NOPASS trial) to assess the effect of pre-operative pazopanib therapy for unselected high-risk STS patients [2].

Pazopanib is a multi-target tyrosine kinase inhibitor with activity against vascular endothelial growth factors 1, 2, and 3, and platelet-derived growth factors. It is approved in the treatment of metastatic or non-resectable STS. In a pivotal phase III trial, which was limited to pre-treated patients with non-adipocytic STS, 6% of patients treated with the drug showed partial response and 67% stable disease, leading to a median progression-free survival of 4.6 months versus 1.6 months in the placebo arm [3]. Activity of pazopanib against liposarcoma [4] and angiosarcoma [5,6] was shown in subsequent studies with response rates of 20% or more. Within the GISG-04 trial, the histopathological diagnosis of STS was proven by biopsy. Afterwards, patients received 21 days of pazopanib therapy followed by surgery. The primary endpoint was metabolic response defined as a >50% decrease in the mean standardized uptake value (SUVmean) in positron emission tomography–computed tomography (PET/CT). One of the secondary endpoints was histopathological regression based on thorough histopathological analyses of the resection specimen. The trial was carried out from 2013 to 2016 and met neither its primary nor its secondary endpoint [7]. Relevant treatment effects were observed in only a single patient out of a total of 21 patients participating in the study [7]. Independent of the study result, neoadjuvantly treated STS resection specimens and pre-therapeutic biopsies were collected and made available for biomarker analyses.

In a recently published study, a few genetic biomarkers showed an association with response to pazopanib in advanced STS patients, making it worthwhile to screen for these biomarkers in our cohort as well [8]. Besides the option of establishing a genetic biomarker for pazopanib treatment, a thorough genetic assessment is increasingly important for STS classification. Recently, it has been speculated that with next generation sequencing (NGS), STS entities will be characterized by recurrent genetic changes and diagnostic accuracy can be improved [9]. It is very likely that the current classification of STS, which is based on the origin from the normal mesenchymal counterpart, will change into a classification based on molecular genetics. 

The abovementioned considerations have prompted us to conduct an NGS study based on molecular profiling of the pre- and post-treatment STS cases in the GISG-04/NOPASS study by applying the Oncomine Focus Assay (Ion Torrent), which is a targeted multi-biomarker assay specific to 52 genes assumed relevant for cancer. It can assess both DNA and RNA, using paraffin tumor samples, for indels, hotspots, gene fusions, single nucleotide variants, and copy number variations.

## 2. Materials and Methods

### 2.1. Patients and Sample Size

The herein presented patient cohort has already been studied, and patient data derived from different analyses have been published elsewhere [2,7,10]. Out of the 21 patients enrolled in the GISG-04/NOPASS trial, tissue for histopathological analyses was available from 18 patients (9 females, 9 males) (Appendix A). In total, 30 tumor samples consisting of 12 pre-therapeutic biopsies and 18 specimens after neoadjuvant pazopanib therapy (16 resection specimens of the primary tumor, two metastases) were available for NGS analyses. For 12 patients, both pre- and post-therapeutic specimens were available for comparison. 

### 2.2. Study Design

Details of the study protocol have been previously published [2]. In brief, patients were ≥18 years of age, had a primary, resectable, non-metastatic, histologically confirmed high-risk (FNCLCC grade 2/3, diameter ≥ 5 cm) STS of any location for which upfront resection was planned. Patients had measurable disease according to Response Evaluation Criteria in Solid Tumors (RECIST) version 1.1 [11]. The treatment regimen consisted of pazopanib 800 mg daily for 21 days as neoadjuvant therapy. An interval of 7–14 days between pazopanib therapy completion and surgery was allowed in order to minimize potential perioperative complications due to pazopanib. The planned sample size of the trial was 35 patients. However, enrollment had to be stopped after 21 patients following a futility analysis that showed the probability of metabolic response in at least 40% of patients, which was the pre-defined outcome, to be below 5%.

### 2.3. Pathological Examination

Pre- and post-therapeutic specimens were histopathologically assessed and diagnosed according to the WHO Classification for Soft Tissue Tumors [12]. The percentage and type (hyaline necrosis, apoptosis, scar tissue, hemorrhagic necrosis) of regression was recorded in analogy to Grabellus et al. [13], as well as data regarding tumor size, resection status, histological subtype, FNCLCC grade (G1–3), and TNM (according to the eighth edition of bone and soft tissue sarcomas classification [14]).

### 2.4. TCGA Data Retrieval and Processing

Clinicopathological and genomic data of the TCGA sarcoma cohort (*n* = 206 patients; [15]) were explored using cBioPortal for Cancer Genomics (http://www.cbioportal.org/ (accessed on 15 February 2021); [16,17]). Genomic alterations for the 52 genes included in the Oncomine Focus Assay were visualized as Oncoprint for all 206 samples, along with the histologic tumor type.

### 2.5. Targeted Next Generation Sequencing and Data Analysis

Tumor areas of formalin-fixed paraffin-embedded (FFPE) samples were marked on hematoxylin and eosin (H&E)-stained slides by a pathologist for DNA and RNA isolation. The corresponding areas were macrodissected from one to three 10 µm sections using the RecoverAll™ Multi-Sample RNA/DNA Isolation Workflow (cat. no. A26135, Invitrogen, Thermo Fisher Scientific, Waltham, MA, USA). Isolated DNA and RNA were quantified using the Qubit dsDNA HS Assay Kit (cat # Q32854, Invitrogen, Thermo Fisher Scientific) or the Qubit RNA HS Assay Kit (cat # Q32855, Invitrogen), respectively. Libraries of FFPE isolated DNA (10 ng) and RNA (10 ng) were prepared by the Ion Chef Instrument (Thermo Fisher Scientific) using the Oncomine Focus Assay (OFA, w2.5, cat # A42008, Ion Torrent, Thermo Fisher Scientific). The OFA covers 35 hotspot genes, 19 copy number variants, and 23 fusion drivers, resulting in a total of 52 genes. Eight libraries were pooled and loaded onto each Ion 520 chip (Thermo Fisher Scientific) by the Ion Chef Instrument (Thermo Fisher Scientific). Sequencing was performed on an Ion GeneStudio S5 Prime System (Thermo Fisher Scientific). The mean read depth for targeted regions was 1880x. Signal processing, base calling, alignment (hg19), and variant calling were carried out using Torrent Suite Software (version 5.12; Thermo Fisher Scientific) followed by annotation and custom/manual filtering using Ion Reporter (version 5.12; Thermo Fisher Scientific). Variants were checked for germline or somatic origin and further evaluated using the Catalogue of Somatic Mutations database (COSMIC v93, https://cancer.sanger.ac.uk/cosmic (accessed on 15 February 2021)) [18], dbSNP (2.0 build 154, NCBI; https://www.ncbi.nlm.nih.gov/projects/SNP/ (accessed on 15 February 2021)), the ExAC/gnomAD browser [19], and ClinVar [20]. Only single or multi-nucleotide variants with a read depth of at least 500× and a variant allele frequency of ≥10% were considered. Copy number variation (CNV) calling was carried out using the CNV algorithm in Ion Reporter. Sequencing reads were visualized using the Integrative Genomics Viewer Browser (IGV 2.9, http://www.broadinstitute. org/igv/ (accessed on 15 February 2021)) [21].

## 3. Results

### 3.1. Pathological Examination

In total, post-therapeutic specimens of 18 patients (16 primary tumors, two metastases) who had undergone neoadjuvant pazopanib treatment were histopathologically evaluable. Median age at first diagnosis of STS was 68.5 years (range 49–89 years). The predominant histological diagnosis group of STS was dedifferentiated liposarcoma (44%; 8/18) followed by undifferentiated (pleomorphic) sarcoma (22%; 4/18). Two leiomyosarcomas (11%; 2/18), two pleomorphic liposarcomas (11%; 2/18), one myxoid liposarcoma (6%; 1/18), and one synovial sarcoma (6%; 1/18) were also included (Table 1). The main tumor location was the retroperitoneum (38%) followed by the extremities (33%), trunk (5%), and inner organs (5%) (Table 1).

After pazopanib therapy, tumor regression was not extensive. All but one sarcoma showed vital tumor masses of ≥50% (Appendix A). The only STS with <50% vital tumor mass (histopathological responder) was an undifferentiated (pleomorphic) sarcoma (70% hyaline necrosis, 30% vital tumor cells) (Appendix A). The major type of regression was hyaline necrosis (66%) followed by a combination of hyaline and hemorrhagic necrosis (22%) (Table 1).

### 3.2. Molecular Pathology

To determine possible genetic alterations, both pre- and post-treatment sarcoma samples were sequenced with the OFA covering clinically relevant hotspot mutation regions and CNVs in 52 genes. Based on the TCGA Adult Soft Tissue Sarcoma cohort (*n* = 206 patients), mainly CNVs could be expected in this gene set, while mutations were rare (Appendix A). Alteration frequencies differed depending on the sarcoma histotype, e.g., *CDK4* amplification in dedifferentiated liposarcomas (DDLS) (Figure 1). In line with TCGA data, the most frequently observed genomic alteration in our STS cohort was the *CDK4* amplification in DDLSs (average copy number 17.8, range 8.27 to 37.2) (Figure 1) [12]. Seven out of eight (88%) patients with DDLS showed *CDK4* amplification and in all but one patient, *CDK4* amplification status was consistent between the pre- and post-treatment specimens. However, in one patient (P02), the *CDK4* amplification was detected in the post-treatment sample (copy number 14.6) but not in the corresponding pre-treatment sample. Further investigation revealed that the amplification of *CDK4*, located on chromosome 12q14.1, was visible in the copy number scatter plot of the pre-treatment sample (Appendix A) but was not called by the in-built CNV algorithm. This is attributed to the borderline elevated mean absolute pairwise distance (MAPD) score of 0.64 (in-built threshold of Ion Reporter is ≤0.5). In other words, above an MAPD of 0.5, Ion Reporter does not perform copy number calling but CNVs can be found in the filtered out variants list (no call reason MAPD > 0.5) and by visual inspection of the copy number scatter plots.

Besides two non-synonymous mutations in *MAP2K1* (p.Q203K) and *BRAF* (p.V600E), which both occurred in pleomorphic (undifferentiated) sarcomas (P06, P11), no further mutations could be detected with the OFA.

## 4. Discussion

The current study was designed to analyze genetic alterations in pre- and post-treatment STS cases to detect diagnostically relevant genetic alterations and to elucidate a potential predictive biomarker for pazopanib [3]. This analysis was performed on samples collected within the phase II window-of-opportunity trial of the German Interdisciplinary Sarcoma Group GISG-04/NOPASS. We applied the Oncomine Focus Assay (Ion Torrent), a multi-biomarker NGS assay, which enables the detection of genetic variants in 52 tumor-relevant genes [22]. The assay was designed to align with genes relevant to FDA-approved oncology drugs and allows the concurrent analysis of DNA and RNA isolated from FFPE samples to simultaneously detect hotspot mutations, single nucleotide variants, indels, CNVs, and gene fusions.

In our study, this approach detected a very high frequency of *CDK4* amplification (88%; 7/8) in the group of DDLS, which is in line with data from the TCGA group, showing that 94% (47/50) of DDLSs showed highly recurrent copy number gains for *CDK4* [15]. Interestingly, in our study, *CDK4* amplification was not detectable in one out of four pre-therapeutic DDLS biopsies, although it is independent of pazopanib treatment and treatment response. After detailed analyses of the sequencing data from the sample, it became obvious that the *CDK4* amplification was present but was not called by the in-built CNV algorithm in Ion Reporter. This example clearly underlines that NGS results need manual control by molecular pathologists before they are called or disregarded.

Our STS cohort showed a low somatic mutation burden, which is also in line with TCGA data reporting that approximately only one mutation is found per Mb STS DNA, with a few genes, such as *ATRX*, *RB1*, and *TP53*, being found as mutated [15]. Since the applied OFA covers only 0.03 Mb of coding DNA, a relatively small number of mutations was expected. Nevertheless, we found two additional potentially druggable mutations in two undifferentiated (pleomorphic) sarcoma patients (11%; 2/18). Besides the abovementioned *CDK4* amplification within DDLS samples, a *MAP2K1* missense mutation (E203K) and a *BRAF* missense mutation (V600E) were traceable. The *MAP2K1* E203K mutation was detected in the pre- and post-therapeutic specimens of an undifferentiated (pleomorphic) sarcoma. While this specific mutation has not been reported to occur in STS to date, other MAP2K1 mutations have been investigated and reported. Some *MAP2K1* mutations have already demonstrated druggability within sarcoma patients. Recently, a histiocytic sarcoma patient with a *MAP2K1* F53L mutation received the MEK inhibitor trametinib and showed a rapid and durable complete response for more than 2 years [23]. Similar publications are lacking for the *MAP2K1* E203K mutation detected here. This mutation has been characterized as constitutively active and achieved high levels of phosphorylated ERK within cell line experiments [24]. However, recent molecular docking and molecular dynamic simulations showed that *MAP2K1* E203K may attenuate the inhibitory effects of trametinib on MEK1, making a clinical benefit in this context unlikely [25]. Thus far, there is no evidence to show if the effect of other MEK inhibitors would be equally attenuated by the mutation.

The other potentially druggable mutation we could identify here was a *BRAF* V600E mutation in another undifferentiated (pleomorphic) sarcoma. Similar to other malignant tumors, *BRAF* mutations are also present in sarcomas but with a very low frequency. Within the TCGA cohort of 206 STS samples, no mutation was detectable in *BRAF* [15]. Thus far, only singular case reports have suggested that BRAF inhibitors in combination with MEK inhibitors (dabrafenib/trametinib) could result in a good response in *BRAF* V600E-mutated sarcoma [26]. In a published “basket” study assessing treatment effects of vemurafenib in *BRAF* V600-mutated non-melanoma malignancies, two out of 122 cases were STS and one patient achieved a partial response to vemurafenib [27].

Although the druggability of the two mutations detected here is of a theoretical nature, since the genetic alterations were neither evaluated at the time of treatment nor the potential treatment administered, the finding still demonstrates that even in less mutagenic tumors like STS and even by applying a less comprehensive NGS panel like the OFA, potentially actionable mutations can be found. Other studies showed that targetable mutations for which specific clinical trials were available can be identified in 40–60% of STS patients by applying larger NGS panels [28,29]. In our cohort and based on the rather small OFA panel, at least two out of 18 patients (11%) would have had the option of an individualized therapy. Our study thus underlines that NGS testing is a powerful technology, which offers a great contribution to personalized and precision medicine even in a “mutation unlikely” entity like STS.

Besides these single mutations, the study also aimed in the direction of establishing a predictive biomarker for pazopanib treatment. However, the likelihood of being successful was very low since our cohort comprised only one responder (P12). This patient’s undifferentiated (pleomorphic) sarcoma did not show any genetic alteration based on the OFA. In the literature, mainly by means of case reports, only a few genetic alterations have been associated with pazopanib response to date. In one STS patient with a complete response to pazopanib, hitherto undescribed somatic mutations, including Fms-related receptor tyrosine kinase 1 (*FLT1*) G38S, platelet-derived growth factor receptor alpha (*PDGFRA*) T83S, and platelet-derived growth factor receptor beta (*PDGFRB*) exon 13 skipping, were detected. Moreover, overexpression of GLI1, which in turn upregulated PDGFRB protein expression and promoted phosphorylation, was found. The latter is supposed to be inhibited by pazopanib in a dose-dependent manner [8]. It was also reported that TP53 status can predict response to pazopanib based on 19 advanced STS cases [30]. However, we did not detect a *PDGFRA* mutation in our responder or in other tumor samples within the study population. Unfortunately, gene sequences for *FLT1*, *GLI1*, *PDGFRB*, and *TP53* are not included in the OFA, making it impossible to detect these alterations with our approach. In a future step, it would be ideal to extend the NGS investigation with a broader panel to include these potentially relevant gene regions.

Our study has some limitations: (i) the main drawback of our analysis was the small number of available tumor samples given that enrollment into the trial was stopped based on the result of a futility analysis [7]; (ii) due to availability, we had to apply a rather small NGS gene panel, possibly missing relevant targets. However, the OFA panel covers—with a few exceptions such as *BRCA1* and *BRCA2*—the majority of clinically relevant genetic alterations and is the NGS panel routinely used in our institution.

## 5. Conclusions

The results of this analysis of STS samples retrieved before and after pazopanib therapy suggest that genetic alterations play a role in STS pathogenesis and pathophysiology, thus underlining the importance of an additional, genetically based STS classification complementing, but not replacing, histomorphologic assessment. *CDK4* amplification was present in all but one dedifferentiated liposarcoma, and two potentially targetable mutations of the BRAF/MAPK pathway were detectable in undifferentiated (pleomorphic) sarcomas. Given the low metabolic response rate in this trial, no predictive biomarker for pazopanib could be established. Nevertheless, our results show the utility of NGS testing of STS even in light of the fact that this entity is rather unlikely to show a large number of mutations.

## Figures and Tables

**Figure 1 biology-10-00639-f001:**
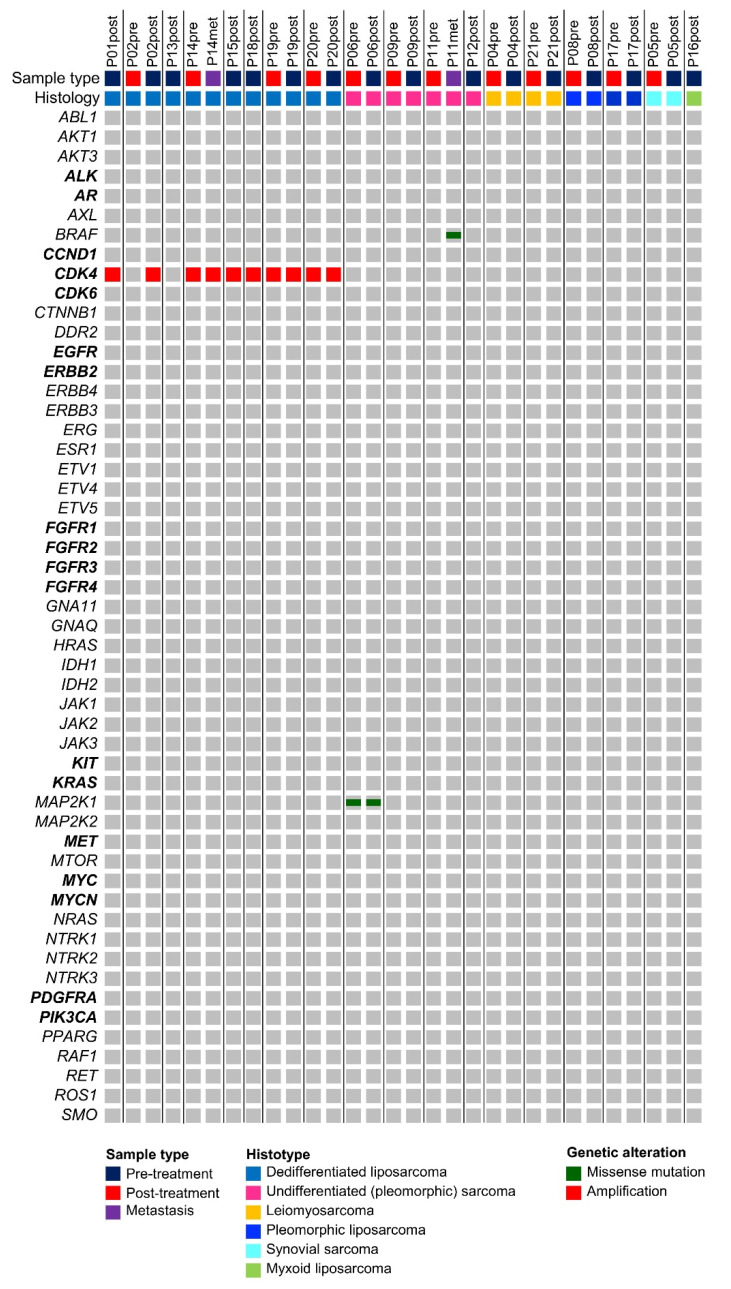
Mutational landscape of STS based on the 52 hotspot, CNV, and fusion genes covered by the Oncomine Focus Assay. All detected variants are shown along the respective STS histotype. As expected, CDK4 amplifications are confined to dedifferentiated liposarcoma. Genes, for which the Oncomine Focus Assay allows the detection of copy number variations, are highlighted in bold.

**Table 1 biology-10-00639-t001:** Clinical and pathological characteristics of the *n* = 18 patients with STS included in the present analysis.

Variable	Number
**Age at first diagnosis, years**	
Median (range)	68.5 (49–89)
**Gender**	
Male	9
Female	9
**Site of primary sarcoma**	
Retroperitoneum	7
Extremities	6
Trunk (superficial)	4
Inner (abdominal) organs	1
**Histologic tumor type**	
Dedifferentiated liposarcoma	8
Undifferentiated (pleomorphic) sarcoma	4
Leiomyosarcoma	2
Pleomorphic liposarcoma	2
Myxoid liposarcoma	1
Synovial sarcoma	1
**Type of necrosis (post-treatment)**	
Hyaline	12
Combined hyaline and hemorrhagic	4
N/A	2

## Data Availability

The data presented in this study are available upon reasonable request from the corresponding author.

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
