# Peer review of "Molecular and Pathological Profiling of Corresponding Treatment-Naïve and Neoadjuvant Pazopanib-Treated High-Risk Soft Tissue Sarcoma Samples of the GISG-04/NOPASS Study"

_biology, 2021, doi:10.3390/biology10070639_

Round 1
Reviewer 1 Report
The authors manage to address the majority of my concerns and the manuscript has improved. The manuscript provides insights on the potential diagnostic potential of genetic profiling for STS patients and would be of interest to readers.
Reviewer 2 Report
No comment.
Reviewer 3 Report
The authors corrected their manuscript according to the comments I had made
This manuscript is a resubmission of an earlier submission. The following is a list of the peer review reports and author responses from that submission.
Round 1
Reviewer 1 Report
Brief Summary:
The aim of the study by Gaiser et al., was to identify molecular biomarkers that could stratify soft tissue sarcoma (STS) patients for better responses to pazopanib, a tyrosine kinase inhibitor approved for the treatment of advanced STS. In addition, the authors aimed at identifying additional genetic mutations that could be used for other target therapeutic approaches in STS. They examined STS tumor tissue from a published clinical trial study (GISG-04/NOPASS), where DNA sequencing data as well as pathohistological characterization were already available. The authors utilized the Oncomine Focus Array to detect genetic variants of 52 genes in tumor samples from 18 STS patients, identifying a prevalent CDK4 amplification in the group of dedifferentiated liposarcoma but no alterations specific to pazopanib response. Moreover, this study identified two potentially druggable MAP2K1 and BRAF mutations in two pleomorphic sarcomas. Although the study did not identify a specific DNA mutation that could predict response to pazopanib, it validates the feasibility of genetic profiling for potentially diagnostic purposes in STS, a tumor entity with relatively low mutational profiles. This manuscript presents interesting data on the diagnostic potential of genetic profiling for STS patients and could serve as the basis of more comprehensive studies to elucidate the impact of next generation sequencing (NGS) on STS prognosis and treatment. However, some minor issues should be addressed for the manuscript to be considered for publication.
Strengths of the study:
- Mutational profiling of STS tumor with established clinicopathological characteristics
- Identification of new genetic alterations in STS
- Revealing the feasibility of NGS approaches for prognostic/diagnostic purposes in STS
Weaknesses of the study:
- Small number of samples may preclude identification of treatment-specific mutations
- Narrow range of NGS used (only 52 genes) may also preclude identification of treatment-specific mutations as well as novel STS subtype-specific mutations
Comments
Simple Summary:
- No need to abbreviate DNA. [minor]
Abstract:
- NGS is not really considered revolutionary anymore, since it is been widely used for a plethora of prognostic and diagnostic purposes in the cancer field. [minor]
Results:
- The authors mention that the GISG-04/NOPASS study did not meet its endpoint criteria but still tumors samples were collected for biomarker analyses. Do the authors have any clinical outcome follow-up information on the cohort of patients from which tumor samples were used in the current study? It would be interesting to assess the relationship between CDK4 amplification and outcome. [minor]
- The authors should more clearly explain how the Oncomine Focus Array sequencing works since it only allows for the detection of copy number variations for only 18/52 genes analyzed. This also limits the significance of the analyses. This could be done either in the Introduction or Results. [minor]
- The labelling of Figure 1 is confusing. To simplify the color-coding I would recommend that the authors remove the row with the Patient IDs since it does not add to the data. [minor]
Discussion:
- The authors should make it clear that the CDK4 amplification identified in their study is independent of pazopanib treatment/response. [minor]
- The authors should discuss in a bit more depth the relevance of their studies the recently published study by Suehara et al., 2020 (PMID: 32567826). [minor]
Conclusions:
- Last sentence is a repetition of the last sentence from the 5th paragraph of the Discussion. The authors should rephrase. [minor]
Reviewer 2 Report
Key word must be improved that the presentation of V600E is nothing must be used the form BRAF V600E that is true presence of the detected gene a its mutation. The same is the presntation of the other genes.
the results are poor to the conclusions. The table is not in correspondence to described results that is presented not clearly. The analysed cohorte is inhomogenous histologicalz that the presenceof histologicall signs can be registred the better chracter of STS. The neccesity of NGS in conclusions is not described in wresults. The relationships between histologz imunohistology and NGS can be improved the results.
Reviewer 3 Report
The authors present a well written paper on NGS in STS patients. Unfortunately the results are not encouraging in the use of NGS for Pazopanib response prediction as could have been expected from the first paper out of this research project with only one responding patient to Pazopanib. As this paper does not significantly add to the current knowledge on NGS detected Mutations that could be draggable for STS patients it is not appropriate for publication in Cancers.
Reviewer 4 Report
In this study, Gaiser and colleagues analyze a small cohort of NSG data from a study of STS of biopsies pre- and post- pazopanib therapy. The primary and secondary outcomes - radiographic and histopathological responses - were not met and the study was halted prematurely due to futility.
Here, the authors report the results of an NSG Oncomine Focus Assay that covers 52 genes at 0.03 Mb of DNA. They include 30 tumors from 18 patients with pre- and post- pazopanib treatment samples included for most patients. This study has many significant limitations, including the small size of the cohort, the shallow coverage of sequencing, and the limited number of genes included on the panel. In fact, the panel did not include PDGFRB and TP53, which have been reported to be associated with response to pazopanib. As a result, the impact of this study is extremely limited. The NSG did detect two potentially targetable mutations in MAP2K1 and BRAF but mutations in both MAP2K1 and BRAF have previously been reported in STS.
While I believe this study has little to no impact, I do commend the authors for being transparent and forthright on the study limitations.
Reviewer 5 Report
The article entitled « Molecular and pathological profiling of corresponding treatment naïve and neoadjuvant Pazopanib treated high-risk soft tissue sarcoma samples of the GISG-04/NOPASS study » is presenting the results of new molecular analyses done on samples from a previously published clinical study. The RNA and DNA of pre-and post treatment samples are sequenced using the Oncomine Focus assay to detect the mutations, copy number variations and genes fusion in 52 druggable genes.
The objective is to search for genetic alterations that might potentially help to improve the classification of soft tissue carcinoma (STS), to check if such molecular testing might be usefull for personalized medicine and to find potential predictive biomarkers of Pazopanib treatment.
The article is very well written, and clear. The techniques used are well described and well chosen. The figures are explicit.
But the results are scarce, and poorly conclusive. And more serious, this was predictable before starting the sequencing analyses. Those main problems are well presented and discussed by the authors, but are nonetheless significant problems:
- The cohort is small (18 patients, 30 samples pre-post treatment)
- Only one patient was responding to treatment
- The soft tissue carcinoma is known to present very few recurrent genetic alterations
The combination of these three issues prevents the original objectives from being achieved, and the authors use a roundabout conclusion (“NGS testing is a powerful and revolutionary technology helping to improve diagnostic accuracy and offering some patients the chance for personalized medicine even in a “mutation unlikely” cohort like STS”).
Minor comments:
The introduction should contain several additional information to help the understanding:
- a short description of the anti-cancer molecular mechanisms of the Pazopanib should be added.
- the global rate of Pazopanib response to treatment in the different clinical settings should be added
- the known recurrent genetic alteration in the soft tissue carcinoma and their frequencies should be added. Precise if the few biomarkers found in the ref 5 are part of the known recurrent genetic alterations in soft tissue carcinoma.
The readability of the table 1, to understand what variable are categories or subcategories, can be improved by adding lines and bold characters (eg: a line above Gender and a line after female, with Gender in bold).
In the supplemental Table S1, please indicate what mean R0 and R1 in the column resection status, which patient was the only responder, and which patients are the 18 for whom the molecular analysis was performed (those of the Table 1).
The authors should moderate the formulation of the conclusion line 273: “genetic alterations play a role in STS pathogenesis and pathophysiology”.